# Enhancing Virtual Try-On with Synthetic Pairs and Error-Aware Noise Scheduling

## Abstract

Given an isolated garment image in a canonical product view and a separate image of a person, the virtual try-on task aims to generate a new image of the person wearing the target garment. Prior virtual try-on works face two major challenges in achieving this goal: a) the paired (human, garment) training data has limited availability; b) generating textures on the human that perfectly match that of the prompted garment is difficult, often resulting in distorted text and faded textures. Our work addresses these issues through a dual approach. First, we introduce a garment extraction model that generates (human, synthetic garment) pairs from a single image of a clothed individual. The synthetic pairs can then be used to augment the training of virtual try-on. Second, we propose an Error-Aware Refinement-based Schrödinger Bridge (EARSB) that surgically targets localized generation errors for correcting the output of a virtual try-on model. To identify likely errors, we propose a weakly-supervised error classifier that localizes regions for refinement, subsequently augmenting the Schrödinger Bridge's noise schedule with its confidence heatmap. Experiments on VITON-HD and DressCode-Upper demonstrate that our synthetic data augmentation enhances the performance of prior work, while EARSB improves the overall image quality. In user studies, our model is preferred by the users in an average of 59% of cases.

## 1 Introduction

Virtual try-on aims to generate a photo-realistic image of a target person wearing a prompted product-view garment (Li et al., 2023; Zhu et al., 2023; Yang et al., 2024). It allows users to visualize how garments would fit and appear on their bodies without the need for physical trials. While recent methods have made significant strides in this field (Shim et al., 2024; Xie et al., 2023; Yang et al., 2024; Kim et al., 2024), noticeable artifacts such as text distortion and faded textures persist in generated images. For example, as illustrated in the second row of Fig. 1, the logo and the text on the t-shirt noticeably fade away in the initial image generated by a prior try-on GAN model (Shim et al., 2024). These imperfections stem from two primary challenges in virtual try-on: limited data availability and the complexity of accurate garment texture deformation. Addressing the issues, we propose a two-pronged approach: augmenting training data through cost-effective synthetic data generation, and surgically targeting known generation artifacts using our proposed Error-Aware Refinement-based Schrödinger Bridge (EARSB).

At a minimum, the training data of virtual try-on requires paired (human, product-view garment) images. The product-view garment image is a canonical, front-facing view of the clothing with a clean background. A substantial amount of data is needed to capture the combinatorial space comprising all possible human poses, skin tones, viewing angles, and their respective physical interactions with fabric textures, shapes, letterings, and other material properties. Unfortunately, these images are generally available only on copyright-protected product webpages and therefore not readily available for use. To mitigate this issue, we propose to augment training with synthetic data generated from the easier symmetric human-to-garment task, wherein we train a garment-extraction model to extract a canonical product-view garment image from an image of a clothed person. This will allow us to create synthetic *paired* training data from unpaired datasets (Liu et al., 2016; Fu et al., 2022; Xie et al., 2021). Our results demonstrate that incorporating the more readily available synthetic training pairs can improve image generation quality in the virtual try-on task.

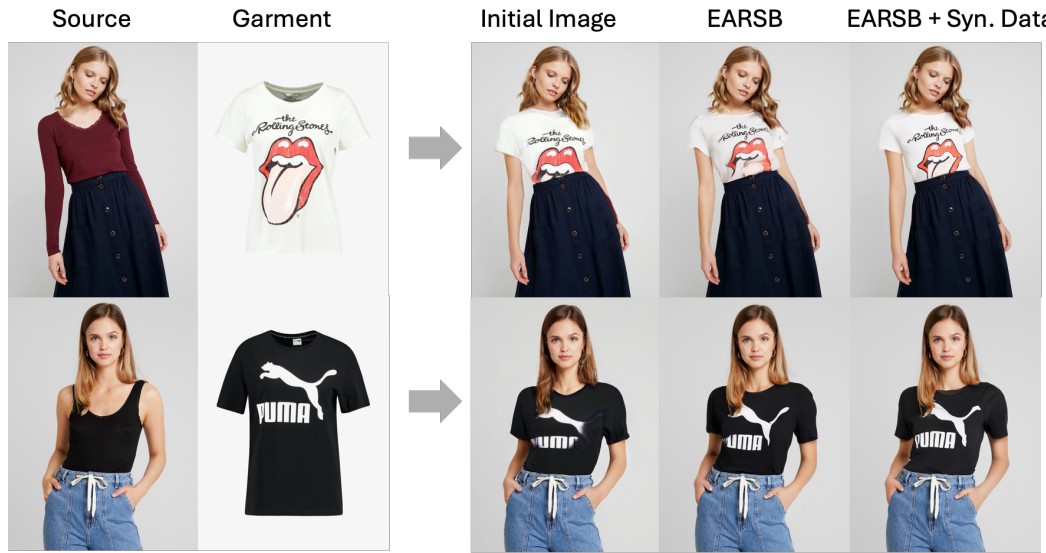

Figure 1: Example of our proposed Error-Aware Refinement SchrödingerBridge EARSB. From left to right is the source human image, the garment image, the initiate try-on image generated by GAN (Shim et al., 2024), the image refined by our EARSB, and the image further refined by EARSB with synthetic data augmentation in training.

In addition to addressing the data scarcity issue, we aim to construct a refinement model that can make localized adjustments to a weaker model's generation results. Our approach draws inspiration from classical boosting approaches where every model in a cascade of models targets the shortcomings of the preceding models. We are interested in a targeted refinement approach for two main reasons. First, it allows a training objective that is focused solely on fixing specific errors. Second, it potentially saves computation when initial predictions are sufficiently good.

Two components are necessary to achieve such a pipeline: a classifier for identifying localized generation errors, and a refinement model that can re-synthesize content specifically in these localized regions. While it may be possible to learn an artifact classifier by repurposing a GAN's fake/real discriminator, we found that an effective weakly-supervised classifier can be constructed with just a few hours of manual labeling of generation errors. For example, we use bounding boxes to quickly annotate the noticeable artifacts in 5% images generated by existing GANs (Lee et al. (2022); Xie et al. (2023); Shim et al. (2024)). Another benefit of this approach is that it can be easily tailored for the errors of a specific model. The resulting weakly-supervised classifier will produce an error map highlighting low-quality regions. Subsequently, we adopt an Image-to-Image Schrodinger Bridge ($I^2SB$) (Liu et al., 2023) to learn the refinement of these regions in the GAN-generated images. While typical diffusion models map from noise to data, $I^2SB$ constructs a SchrödingerBridge (SB) that allows us to map from data to data, or in our setup, generations with artifacts to ground truth images. We direct the SB process to focus on the localized errors by incorporating the classifier's prediction error into the noise schedule, which we describe in more detail in Sec. 3.2.1. In the second row of Fig. 1, our refinement SB model (*i.e.*, EARSB) corrects the faded texture in the initial GAN-generated image.

The contributions of our paper are summarized as follows:

- We introduce (human, synthetic garment) pairs as an augmentation in the training of virtual try-on task. The synthetic garment is obtained from our human-to-garment model, which can generate product-view garment images from human images.
- We propose EARSB, an Error-Aware Refinement SB with a weakly-supervised noise schedule. It improves the low-quality region of a GAN-generated image based on an adaptive noise schedule that targets known artifacts.
- Extensive experiments on two datasets (VITON-HD (Lee et al., 2022) and DressCode-Upper (Morelli et al., 2022)) show that EARSB enhances the quality of the images generated by prior work, and is preferred by the users in 59% cases on average.

## 2 RELATED WORK

**Training with Synthetic Data.** The addition of synthetic data is often an effective means of improving downstream task performance when it is difficult to amass real data at the necessary scale. This has been demonstrated in the domains of image generation (Karras et al., 2020; Shivashankar & Miller, 2023), classification tasks (Sarıyıldız et al., 2023), and image editing (Brooks et al., 2023; Wasserman et al., 2024). Careful applications can also be used to ameliorate dataset imbalance issues, as shown in Dablain et al. (2022). Other works such as Alemohammad et al. (2024) use self-synthesized data to provide negative guidance for the diffusion model. Our incorporation of synthetic data in the virtual try-on task tackles a specific sub-problem in the broader image editing domain and is similar in spirit to Brooks et al. (2023); Wasserman et al. (2024). Specifically, we aim to synthesize paired training data that satisfies the stringent requirements of virtual try-on paired training data – a canonical product-view garment image paired with an example of it being worn. Images of people in clothing are readily available, but it is difficult to obtain a product-view image of the exact clothing they are wearing. To address this, our work proposes and tackles the human-to-garment, which is roughly symmetric to the virtual try-on task and aims to extract the clothing from a person's photo and project it to the canonical product view.

**Virtual Try-On.** There has been a shift from earlier GAN-based framework (Han et al., 2019; Lee et al., 2022; Shim et al., 2024; Xie et al., 2023; Li et al., 2023) to diffusion-based methods (Kim et al., 2024) in the virtual try-on literature. Diffusion models fit an SDE process mapping from the image distribution to the noise distribution, and tend to be easier to train than GAN-based approaches due to the simplicity of the L2 denoising loss (Gao et al., 2023; Dhariwal & Nichol, 2021; Song & Ermon, 2019). At inference, the diffusion model denoises a random Gaussian noise distribution to a human-readable image via multiple sampling steps. Yang et al. (2024) proposes a parameter-efficient approach that concatenates the human image and the garment images along the spatial dimension such that the self-attention layer in the denoising UNet can achieve texture transfer without extra parameters. In (Kim et al., 2024), the authors introduce additional cross-attention layers to learn the semantic correspondences between the garment and the human image. The methods in (Ning et al., 2024; Li et al., 2024; Baldrati et al., 2023) align different embedding spaces in the attention module to achieve flexible clothing editing after try-on, such as style change or graphics insertion. In contrast to prior work that samples from random noise, we build upon recent advances in Schrödinger bridges, notably Liu et al. (2023), to directly sample from an initial image generated by a cheaper, but reasonably good GAN model. Our work is similar in spirit to Zeng et al. (2024), which initializes the noisy image with a GAN-generated image and small amounts of random noise. However, we consider the Schrödinger bridge formulation to be a more principled approach to mapping from a pre-existing result. Our work also explores varying the local noise schedule based on how much we expect the original result to be modified at a given location.

## 3 METHOD

Virtual try-on aims to re-synthesize a given human image with a newly specified garment. Critically, we wish to maintain the original pose of the human model, while plausibly depicting how the specified garment would deform and appear when worn on said model's body. We study the feasibility of improving current virtual try-on models by addressing both the data scarcity issue as well as through a refinement model designed to find-and-fix localized generation errors. We describe our approach to generating synthetic training data in 3.1, and then our EARSB refinement model in 3.2.

### 3.1 AUGMENTED TRAINING WITH SYNTHETIC PAIRED DATA

The ideal training setup for a virtual try-on model comes in the form of human-garment image pairs, where we have both the ground truth image of a human model wearing a target garment, as well as a canonical product-view image of the isolated garment. We will refer to this setup as paired for short. Given these data, one can then train a try-on model by masking out the target garment from the ground-truth model's image and reconstructing it using the product-view garment image (see the left-most preprocessing step of 3).

However, acquiring a substantial volume of such paired data is challenging due to copyright and brand protection. Instead, obtaining single human images is considerably more feasible (Liu et al.,

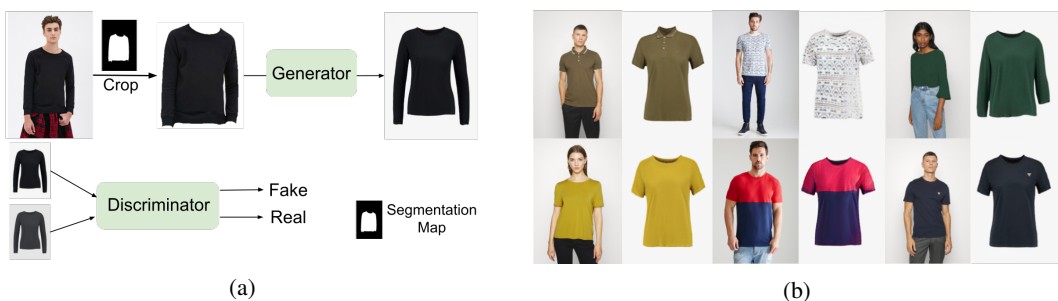

(a)                                 (b)

Figure 2: (a) Our human-to-garment model, which is explained in Sec. 3.1.1 (b) Examples of the constructed (human, synthetic garment) pairs.

2016; Fu et al., 2022; Xie et al., 2021; Li et al., 2024). This observation motivates our proposal of the human-to-garment task, which enables us to easily obtain (real human, synthetic garment) pairs from single human images. These pairs can subsequently be utilized to augment the training data for our garment-on-human task (*i.e.*, virtual try-on). In the following, we explain the architecture of our human-to-garment model and how it is used to augment the virtual try-on training.

### 3.1.1 HUMAN-TO-GARMENT MODEL

While virtual try-on requires generating skin and deforming the product-view garment to accommodate diverse postures, the human-to-garment task simply aims to map the clothing item to its canonical view. To achieve this, we extract the clothing on the person using its segmentation map, and feed it to a GAN generator that generates a canonical view of the clothing, as illustrated in Fig. 2a. The generator architecture is based on the UNet model proposed in (Han et al., 2019), which uses a flow-like mechanism for warping latent features in an optical-flow-like manner. The generator was trained using a combined L1 reconstruction and adversarial loss.

**Generating Synthetic Data H2G-UH and H2G-FH.** With the above human-to-garment model, we can obtain the product-view garments from single-human images (Liu et al., 2016; Fu et al., 2022; Xie et al., 2021), and create (real human, synthetic garment) pairs to facilitate the try-on task. We use the following criteria to ensure quality: a) The human image has a clean background (low pixel variance in the non-human region); b) The human image is of front view (classified by its DensePose representation (Güler et al., 2018)); c) the reconstruction error (LPIPS distance) is small when reconstructing the human image in a try-on model using the (human, synthetic garment) pair (*e.g.*, (Lee et al., 2022; Shim et al., 2024)). Under these criteria, we select human images from DeepFashion2 (Liu et al., 2016) and UPT (Xie et al., 2021), eventually creating 12,730 synthetic pairs of upper-body human images (referred to as H2G-UH) and 8,939 pairs of full-body human images (referred to as H2G-FH). Examples of the synthetic pairs are shown in Fig. 2b.

### 3.1.2 VIRTUAL TRY-ON (GARMENT-ON-HUMAN) TRAINING

Even with our filtering criteria, our synthetic data remains imperfect. We explore two means of limiting the effect of the real-synthetic domain gap: (a) pretraining the try-on model using synthetic pairs and finetune it on real pairs (Kumar et al., 2022); (b) training simultaneously on real and synthetic data, but conditioning the try-on model on a real/synthetic flag, similarly to (Jun et al., 2020). The difference between these two is that in the first augmentation, the model only sees synthetic data at an early training stage, while in the second augmentation, the model takes both real and synthetic pairs throughout the training. We found empirically that the second augmentation performs slightly better than the first (See Sec. 4.1).

### 3.2 ERROR-AWARE REFINEMENT SCHRÖDINGER BRIDGE

Our refinement objective is weakly inspired by boosting methods in that we wish to fit a *targeted* refinement model that is trained specifically on the generation errors of an existing model. Let $x_1$ be a virtual try-on rendering from an initial lightweight GAN-based model (Lee et al., 2022; Shim et al., 2024), and $x_0$ is the ground-truth target image. We wish to construct a refinement model that maps

Figure 3: The diffusion process in our EARSB. We first use a try-on GAN model that takes the masked human image $\bar{x}_0$, its pose representation $P$, and its garment $C$ as input to generate an initial human image $x_1$. $x_1$ is fed to our weakly-supervised classifier to obtain the error map $M$ (see Sec. 3.2.1). This map reweights the noise distribution $\epsilon$ to $\epsilon^r$ in I$^2$SB diffusion (see Sec. 3.2.2).

from $x_1$ to $x_0$. (See bottom right of Fig.3). To achieve this, we adopt a solution based on diffusion Schrödinger bridges as formulated in I$^2$SB (Liu et al., 2023). I$^2$SB constructs a SchrödingerBridge (SB) that allows us to map from data to data in accordance with our goals.

A naïvely trained Schrödinger bridge model is provided training pairs $(x_0, x_1)$ to map between, but is not explicitly aware of which locations should from $x_1$ require alteration. It would have to learn this implicitly from its training. We propose a more explicit approach where we identify the errors in $x_1$ with a confidence map $M$. As shown in Fig. 3, $M$ reweights the noise schedule of the I$^2$SB stochastic process and assigns a higher volume of noise to the highlighted low-quality regions. At the same time, an area assigned a low noise volume (low confidence of error) will have reduced capacity for altering the original pixel values.

### 3.2.1 OBTAINING THE ERROR MAP

The error map $M$ is a score map that highlights the corrupted or incorrect area of the initial image $x_1$. While it may be possible to learn an artifact classifier by repurposing a GAN's fake/real discriminator, we found that an effective weakly-supervised classifier can be constructed with just a few hours of manual labeling of generation errors. As shown in Fig. 3, our weakly-supervised classifier consists of two encoders with a cross-attention module. The two encoder outputs are fused with cross attention to output a sigmoid-activated error map.

Ideally, the classifier should predict 1 for GAN-generated low-quality regions and 0 for error-free regions in $M$. However, we do not have full annotations for where all the generated artifacts are located. Further, automated metrics such as localized reconstruction errors would produce too many false positives, as they do not account for the full distribution of valid renderings. To mitigate this issue, we hand-labeled 5% GAN-generated images in the training set at the patch-level for poorly generated regions. We then train with a weakly supervised objective that uses patch-level supervision for labeled images, and image-level supervision (GAN-generated vs real) for unlabeled images.

Let $x_0, x_1^u, x_1^l$ be the real human image, the unlabeled GAN-generated human image, and the labeled GAN-generated image with bounding boxes annotating artifacts. Our loss terms are defined as:

$$\mathcal{L}_{img} = -\log\left(\text{WSC}(x_1^u, C)_{\max}\right) + \log\left(1 - \text{WSC}(x_0, C)_{\max}\right) \tag{1}$$

$$\mathcal{L}_{pat} = -\log\left(\text{WSC}(x_1^l, C) \odot B_{box}\right) - \log\left(1 - \text{WSC}(x_1^l, C) \odot (1 - B_{box})\right) \tag{2}$$

where $\mathcal{L}_{img}$ is the image-level loss and $\mathcal{L}_{pat}$ is the patch-level loss. In $\mathcal{L}_{img}$, $\text{WSC}(\cdot)$ is the output error map and $\text{WSC}(\cdot)_{\max}$ denotes the spatially max-pooled score in the error map. This can also be interpreted as a multi-instance learning problem where the training model only knows that at least one region within the image is of low quality. The patch-level loss $\mathcal{L}_{pat}$ further introduces positive

and negative pixel samples and is only available for fully annotated samples. $B_{box}$ is the spatial binary mask for the annotated regions, thereby maximizing and minimizing the scores for regions within and outside of the annotated boxes respectively. Our final loss is: $\mathcal{L}_{\text{WSC}} = \mathcal{L}_{ins} + \mathcal{L}_{pat}$. In practice, our classifier operates and averages results over 3 different feature map resolutions, with additional upsampling for lower resolutions to ensure the same output dimensions.

### 3.2.2 Error-Map Reweighted Schrödinger Bridge Formulation

**Preliminary.** We built upon I$^2$SB, which fits direct mappings between data domains. We define our two domains $x_0 \in X_0$ and $x_1 \in X_1$ as ground truth renderings and model-predicted renderings respectively. The corresponding diffusion process in I$^2$SB is given as:

$$x_t = \mu_t(x_0, x_1) + \sqrt{\Sigma_t} \cdot \epsilon$$
$$\mu_t = \frac{\bar{\sigma}_t^2}{\bar{\sigma}_t^2 + \sigma_t^2} x_0 + \frac{\sigma_t^2}{\bar{\sigma}_t^2 + \sigma_t^2} x_1, \ \Sigma_t = \frac{\bar{\sigma}_t^2 \sigma_t^2}{\bar{\sigma}_t^2 + \sigma_t^2} \cdot I,$$

(3)

where $\sigma_t^2 = \int_0^t \beta_\tau d\tau$, $\bar{\sigma}_t^2 = \int_t^1 \beta_\tau d\tau$ and $\beta_\tau \in \mathcal{R}$ is the chosen noise schedule. $\epsilon \sim \mathcal{N}(0, I)$ is a random Gaussian noise. The above equation shows that $x_t$ can be obtained by adding the Gaussian noise $\epsilon$ to the interpolation of $x_0$ and $x_1$.

**Error-Map Reweighted Diffusion Process.** Our refinement framework defines $x_1$ to be an initial human image generated by a try-on GAN model (Lee et al., 2022; Shim et al., 2024). Noise schedules in a diffusion process typically vary only with the time variable $t$. We introduce a formulation where the noise level also varies spatially – reweighting the noise distribution in I$^2$SB using the error map $M$ we obtained from the classifier in Sec. 3.2.1. Good regions will be assigned less noise (*i.e.*, smaller variance) in the diffusion process, while poor quality regions will be assigned more:

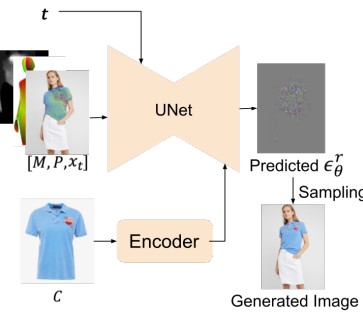

$$x_t = \mu_t(x_0, x_1) + \sqrt{\Sigma_t} \cdot \epsilon^r,$$ (4)
$$\epsilon^r = M \cdot \epsilon, M = \text{WSC}(x_1, C)$$ (5)

where $\mu_t$ is the same as in Eq. 3 and $\epsilon^r$ is the reweighted noise distribution. $M$ highlights low-quality regions in $x_1$ and assigns higher variance to them in the diffusing process, thus steering the model to learn rich textures with high fidelity.

Figure 4: EARSB's denoising UNet. Inputs are the concatenation of $M$, $P$ and $x_1$; outputs are the predicted reweighted noise $\epsilon_\theta^r$.

**Sampling Process.** Different from prior soft-attention-based UNets (Zeng et al., 2024; Yang et al., 2024; Kim et al., 2024; Morelli et al., 2023), our model adopts a UNet architecture with cloth-flow-learning modules to predict the adaptive noise distribution for more precise garment deformation (Han et al., 2019). See Appendix B for the detailed model architecture. As shown in Fig. 4, it encodes the concatenation of the error map $M$, the pose representation $P$ and the noisy image $x_t$. A separate encoder is used to learn the feature representation of the given garment $C$. The output noise distribution of the UNet at time $t$ is $\epsilon_\theta^r(\cdot, t)$, where $(\cdot, t)$ omits the inputs $M, P, x_1, C$. With $\epsilon_\theta^r(\cdot, t)$, we define our sampling process:

$$\hat{x}_0 = x_t - \sqrt{\Sigma_t} \cdot \epsilon_\theta^r(\cdot, t)$$ (6)

$$x_{t-\Delta t} = \hat{\mu}_{t-\Delta t}(\hat{x}_0, x_t) + M \cdot \sqrt{\hat{\Sigma}_t} \cdot \epsilon$$ (7)

$$\hat{\mu}_{t-\Delta t} = \frac{\sigma_{t-\Delta t}^2}{\sigma_t^2} \hat{x}_0 + \frac{\sigma_t^2 - \sigma_{t-\Delta t}^2}{\sigma_t^2} x_t, \ \hat{\Sigma}_t = \frac{\sigma_{t-\Delta t}^2(\sigma_t^2 - \sigma_{t-\Delta t}^2)}{\sigma_t^2}$$ (8)

where $\Delta t > 0$ and it is the sampling interval. Starting from $t = 1$, the process iteratively refines the initial GAN-generated human image $x_1$ with the error map $M$. When $M$ is all ones, our model reverts to the I$^2$SB formulation. When $M$ is all zeros (*i.e.*, no error), the initial image $x_1$ is believed to be perfect so our UNet should predict zero noise in $\epsilon_\theta^r$ and $x_1$ does not need to be refined in the sampling process.

The training objective of our model is the mean squared error between the predicted noise $\epsilon_\theta^r$ and the reweighted Gaussian noise $\epsilon^r$

$$\mathcal{L}_{\text{EARSB}} = ||\epsilon_\theta^r([x_1; P; M], C, t) - \epsilon^r||^2 \tag{9}$$

### 3.2.3 Further Improvements via Classifier Guidance and Expert Denoisers

Classifier guidance has been demonstrated to improve the image fidelity in diffusion models where the classifier often refers to an object category classifier (Dhariwal & Nichol, 2021). In our formulation of the weakly-supervised classifier WSC, the classifier guidance should give a direction toward the real data distribution. (Chung et al., 2023) shows that we can estimate the guidance score $\nabla_{x_t} \log p(y|x_t)$ using the predicted clean sample $\hat{x}_0$: $\nabla_{x_t} \log p(y|x_t) \simeq \nabla_{x_t} \log p(y|\hat{x}_0)$, where $y$ is the class label. Since the label for real data is 0 in WSC, the classifier guidance gives us

$$\hat{\mu}_{t-\Delta t} \leftarrow \hat{\mu}_{t-\Delta t} + M \cdot \hat{\Sigma}_t \cdot \nabla_{x_t} \log p(\mathbf{0}|\hat{x}_0) \tag{10}$$

where $p(\mathbf{0}|\hat{x}_0) = 1 - \text{WSC}(\hat{x}_0, C)$. We found scaling the guidance score to be larger and clamping its value (*e.g.*, $[-0.1, 0.1]$) helps strengthen the guidance without image quality loss.

Further improvements are made via expert denoisers (Balaji et al., 2022), wherein the trained EARSB model is then split into two models, respectively fine-tuned on denoising ranges $t \in [0, 0.5]$ and $t \in [0.5, 1]$. We default to this in all variants of our method in the next section as we find it improves results with no trade-offs (other than model size).

## 4 Experiments

**Datasets.** We use VITON-HD, DressCode-Upper, and our synthetic H2G-UH and H2G-FH for training. DressCode-Upper takes the upper clothing category from the original DressCode dataset. They include 11,647, 13,564, 12,730, 8,939 training images, respectively. For synthetic data augmentation, we combine VITON-HD (Lee et al., 2022) with our H2G-UH since both of them include mostly upper-body human images. DressCode-Upper is combined with H2G-FH as both consist of full-body human photos. For evaluation, VITON-HD contains 2,032 (human, garment) test pairs and DressCode-Upper has 1,800 test pairs. Evaluations are conducted under both paired and unpaired settings. In the paired setting, the input garment image and the garment in the human image are the same item. Conversely, a different garment image is selected as input in the unpaired setting.

**Metrics.** We use Structural Similarity Index Measure (SSIM) (Wang et al., 2004), Frechet Inception Distance (FID) (Heusel et al., 2017), Kernel Inception Distance (KID) (Bińkowski et al., 2018), and Learned Perceptual Image Patch Similarity (LPIPS) (Zhang et al., 2018) to evaluate image quality. All the compared methods use the same image size and padding when computing the above metrics.

### 4.1 Comparison with Existing Methods

We compare our EARSB with GAN-based methods such as HR-VTON (Lee et al., 2022), SD-VTON(Shim et al., 2024) and GP-VTON(Xie et al., 2023), as well as Stable Diffusion (SD) (Rombach et al., 2022) based methods including CAT-DM (Zeng et al., 2024), Stable-VTON (Kim et al., 2024), TPD (Yang et al., 2024) and LaDIVTON (Morelli et al., 2023). We control the number of sampling steps by fixing it to 25 in Tab. 1 for all the diffusion methods. The results under different sampling steps on VITON-HD are presented in Fig. 5.

In Tab. 1, EARSB is our model trained without synthetic data augmentation; EARSB+H2G-UH/FH trains with both H2G-UH and H2G-FH. We add the upper-body synthetic subset H2G-UH for the upper-body-human dataset VITON-HD, and the full-body synthetic H2G-FH when on DressCode-Upper. Our refinement model improves all the metrics of the GAN-based methods on both datasets, with further improvements from incorporating synthetic training pairs.

Note that all the compared SD-based methods in Tab. 1 use pretrained SD model weights, which are obtained by training on millions of (image, text) pairs, while our EARSB+H2G-UH/FH was trained from scratch on try-on datasets with only around 24k (human, garment) pairs. Still, EARSB+H2G-UH/FH achieves comparable performance on VITON-HD and even outperforms SD baselines on DressCode-Upper. We also experimented with adding our synthetic data into the training of SD-based methods in Sec. 4.2 and observed similar improvements.

| | VITON-HD | | | | | | DressCode-Upper | | | | | |
| | Unpaired | | Paired | | | | Unpaired | | Paired | | | |
| | FID↓ | KID↓ | FID↓ | KID↓ | SSIM↑ | LPIPS↓ | FID↓ | KID↓ | FID↓ | KID↓ | SSIM↑ | LPIPS↓ |
| **GAN-Based** | | | | | | | | | | | | |
| HR-VTON (Lee et al., 2022) | 10.75 | 0.28 | 8.46 | 0.26 | 0.901 | 0.075 | 15.26 | 0.39 | 11.76 | 0.32 | 0.947 | 0.046 |
| SD-VTON (Shim et al., 2024) | 9.05 | 0.12 | 6.47 | 0.09 | 0.907 | 0.070 | 14.73 | 0.32 | 10.99 | 0.24 | 0.947 | 0.042 |
| GP-VTON (Xie et al., 2023) | 8.61 | 0.86 | 5.53 | 0.07 | 0.913 | 0.064 | 26.19 | 1.71 | 23.66 | 1.59 | 0.816 | 0.262 |
| **SD-Based** | | | | | | | | | | | | |
| LaDI (Morelli et al., 2023) | | | - | | | | 14.88 | 0.39 | 11.61 | 0.32 | 0.939 | 0.057 |
| CAT-DM (Zeng et al., 2024) | 8.55 | 0.10 | 5.98 | 0.07 | 0.908 | 0.067 | 12.91 | 0.29 | 8.58 | 0.16 | 0.948 | 0.038 |
| TPD (Yang et al., 2024) | 8.23 | **0.06** | **4.86** | **0.04** | 0.917 | **0.057** | | | - | | | |
| Stable (Kim et al., 2024) | **8.20** | 0.07 | 5.16 | 0.05 | 0.917 | **0.057** | | | - | | | |
| EARSB | 8.42 | 0.07 | 5.25 | 0.05 | 0.918 | 0.059 | 10.89 | 0.13 | 7.15 | 0.13 | 0.961 | 0.028 |
| EARSB +H2G-UH/FH | 8.26 | **0.06** | 5.14 | **0.04** | **0.919** | 0.058 | **10.70** | **0.11** | **7.05** | **0.11** | **0.965** | **0.026** |

Table 1: Results on VITON-HD (Lee et al., 2022) and DressCode-Upper (Morelli et al., 2022) KID is multiplied by 100. Results of the diffusion methods are reported at 25 sampling steps.

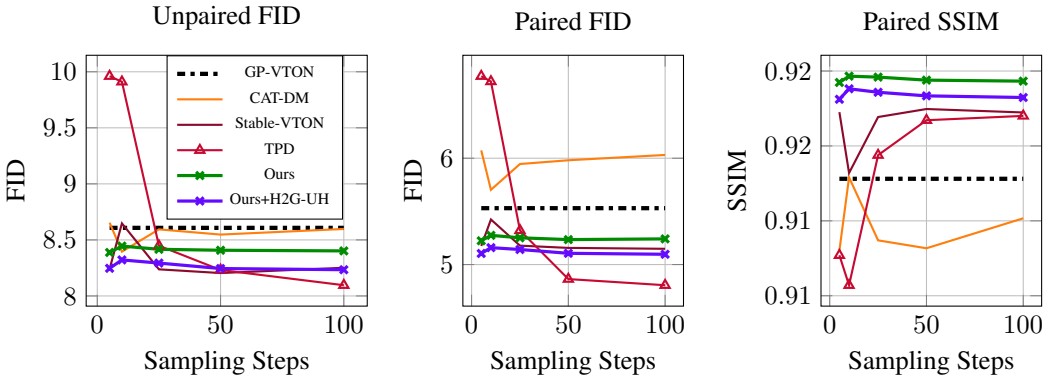

Figure 5: Results on VITON-HD at 5, 10, 25, 50, and 100 sampling steps. Our method consistently improves our baseline starting model GP-VTON (black, dotted line), making it competitive with Stable-VTON (especially at under 50 sampling steps). Legend is shared for all plots.

| Methods | GP-GAN | EARSB | Stable-VTON | EARSB |
|---|---|---|---|---|
| Consistency | 42% | 58% | 38% | 62% |
| Fidelity | 39% | 61% | 45% | 55% |

Table 2: User studies on VITON-HD. Our EARSB is preferred in an average of 59% cases.

**Trade-off Between Sampling Efficiency and Image Quality.** The generated images of diffusion models often show degraded quality with fewer sampling steps. EARSB is based on Schrödinger bridges and inherits the ability to produce reasonable results at fewer step counts as previously demonstrated in (Liu et al., 2023; Lipman et al., 2023). Furthermore, our noise schedule manipulation can keep the correct clothing textures in the GAN-generated image untouched with zero noise level and only fix the erroneous parts, potentially resulting in less image quality degradation with few steps. In Fig. 5, While other SD-based methods have a sharp performance drop with decreasing sampling steps, EARSB and EARSB +H2G-UH show consistent performance across different sampling steps, demonstrating a better trade-off between image quality and computational efficiency.

**User Study.** Amazon MTurk workers were asked to compare the image quality on two aspects: texture consistency, and image fidelity, comparing our model against GP-GAN and Stable-VTON. We randomly selected 100 pairs from VITON-HD to evaluate on, assigning at least 3 workers per image. Tab. 2 reports the study results, where our method is preferred at least 10% more than the GAN-based GP-VTON and the SD-based Stable-VTON (59% overall).

Figure 6: Visualizations on VITON-HD (top row) and DressCode (bottom row). Our EARSB and EARSB+H2G-UH better recover the intricate textures in the garment.

| | | Unpaired | | Paired | | | |
|---|---|---|---|---|---|---|---|
| | | FID↓ | KID↓ | FID↓ | KID↓ | SSIM↑ | LPIPS↓ |
| **(a)** | CAT-DM (Zeng et al., 2024) | 8.56 | 0.10 | 5.90 | 0.06 | 0.911 | 0.067 |
| | CAT-DM +$W$(H2G-UH) | 8.73 | 0.12 | 6.12 | 0.08 | 0.906 | 0.071 |
| | CAT-DM +H2G-UH | 8.36 | 0.09 | 5.67 | 0.06 | 0.913 | 0.063 |
| | Stable-VTON (Kim et al., 2024) | 8.25 | 0.07 | 5.15 | 0.05 | 0.917 | 0.056 |
| | Stable-VTON +$W$(H2G-UH) | 8.47 | 0.08 | 5.32 | 0.06 | 0.910 | 0.060 |
| | Stable-VTON +H2G-UH | **8.17** | 0.07 | **5.04** | 0.05 | **0.919** | **0.054** |
| **(b)** | EARSB | 8.42 | 0.07 | 5.25 | 0.05 | 0.918 | 0.059 |
| | EARSB +$W$(H2G-UH) | 8.68 | 0.08 | 5.44 | 0.06 | 0.909 | 0.063 |
| | EARSB + plain H2G-UH | 9.64 | 0.15 | 6.52 | 0.11 | 0.902 | 0.073 |
| | EARSB pre. H2G-UH | 8.35 | 0.07 | 5.18 | 0.05 | 0.918 | 0.059 |
| | EARSB +H2G-UH | 8.26 | **0.06** | 5.14 | **0.04** | **0.919** | 0.058 |
| **(c)** | EARSB (w.o. $M$) | 9.21 | 0.14 | 6.27 | 0.11 | 0.912 | 0.061 |
| | EARSB (rand($M$)) | 9.13 | 0.16 | 6.55 | 0.11 | 0.902 | 0.071 |
| | EARSB (w.o. CG) | 8.48 | 0.08 | 5.32 | 0.06 | 0.918 | 0.059 |

Table 3: Results of ablations on VITON-HD.

**Qualitative Results.** Fig. 6 gives examples of the generated images using different approaches. The top row is from VITON-HD dataset and the bottom row is from DressCode-Upper. The third images in the two rows are GAN-generated results. We see that our EARSB+H2G-UH/FH in the last column improves the low-quality textures from the GAN-generated images, which are the distorted graphics in the center. More visualized examples can be found in Appendix F.

## 4.2 ABLATIONS

**Synthetic Pairs Augmentation.** To validate the effectiveness of synthetic pairs on enhancing existing diffusion methods, we incorporate H2G-UH into the training of Stable-VTON (Kim et al., 2024) and CAT-DM (Zeng et al., 2024) on the VITON-HD dataset. Here, we use the number of sampling steps as originally published: 2 for CAT-DM, 50 for Stable-VTON, and 25 for our own EARSB. Tab. 3(a) gives the results, with +H2G-UH indicating the data augmentation, indicating that training with our synthetic H2G-UH improves most metrics of the above SD-based models.

To determine the importance of the canonical product-view projection, we replace the synthetic garment in each pair with a randomly warped version of the clothing cropped from the real human image, denoted as +$W$(H2G-UH). The results show that while +$W$(H2G-UH) hinders the performance of all baseline approaches (*i.e.*, CAT-DM, Stable-VTON and EARSB), incorporating H2G-UH improves most metrics. This demonstrates that the synthetic product-view contributes significantly to the observed improvements.

Tab. 3(b) explores different ways of incorporating the synthetic data into training. EARSB pre. H2G-UH is pretrained using the synthetic pairs and finetuned on real data, EARSB+plain H2G-UH is trained using the mixed distribution of the real and synthetic pairs *without* the augmentation label identifying them, and EARSB+H2G-UH uses the mixed data with the identifying label. EARSB pre. H2G-UH is slightly worse than EARSB+H2G-UH. Further, we see the effect of conditioning on the augmentation label, as removing it in EARSB+plain H2G-UH greatly degrades the image quality and causes a significant drop in all metrics.

**Error-Aware Noise Schedule.** The error map adapts the noise distribution according to the quality of the image patches in the initial image $x_1$. This adaptive approach contrasts with a uniform Gaussian noise application across all locations, which would essentially reduce our model to $I^2SB$. As demonstrated in Table 3(c), removing the error map during training (EARSB w.o. $M$) results in a substantial decline across all metrics. Furthermore, we explored using a random error map during the sampling process (EARSB(rand($M$))), which also leads to diminished performance. These results underscore the importance of a meaningful error map in precisely locating and enhancing targeted regions. Additionally, the slight performance degradation observed when removing the classifier guidance (EARSB w.o. CG) suggests that the error map employed in our classifier guidance also contributes to overall image quality improvement. Collectively, these findings highlight the crucial role of our adaptive noise schedule in achieving superior results.

**Validating the Classifier.** Our classifier is weakly supervised by manual artifact annotations and is able to highlight low-quality regions in the initial image $x_1$ with only 5% of the dataset labeled. To validate this, we train two ablations of the weakly-supervised classifier (WSC): the Unsupervised Classifier (UC) that only uses image labels (*i.e.*, fake or real), and the Fake/Real Composite Classifier (CC). CC uses both image fake/real labels as well as fake region-level labels which are created by compositing real image patches and fake image patches. The compositing is a fully automatic alternative to manual labeling that provides patch-level labels. We annotated 100 images in the test set to validate their effectiveness. Fig. 7 shows the pixel-level precision-recall curve for retrieving annotated artifact pixels within the bounding boxes using the classifiers' confidence maps. It is clear that weak supervision remains an incredibly cost-effective approach.

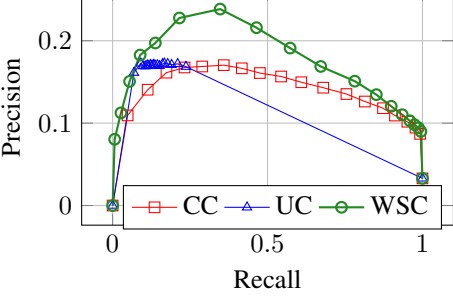

Figure 7: The precision-recall curve for retrieving annotated errors at the pixel level, comparing our weakly-supervised approach to two fully unsupervised baselines (UC, CC). The weakly-supervised classifier (WSC) performs best at retrieving generation artifacts at a nominal labeling cost.

**Effect of Quality of the Initial Image.** Following CAT-DM (Zeng et al., 2024), we use different try-on GAN models, including HR-VTON (Lee et al., 2022), SD-VTON (Shim et al., 2024) and GP-VTON (Xie et al., 2023) at sampling to see how the quality of $x_1$ affects the sampled images at 25 sampling steps. See results in Appendix D, Tab. 4. We observe that: a) our EARSB can refine the GAN-generated image over the GAN baseline; b) the quality of the initial image $x_1$ is positively correlated with the quality of the sampled $\hat{x}_0$; c) our model achieves higher gains over CAT-DM, which also tries to refine the GAN-generated image but without error-aware noise schedule.

## 5 CONCLUSIONS

In this paper, addressing the data limitation in virtual try-on, we introduce a human-to-garment model that generates (human, synthetic garment) pairs from a single image of a clothed individual. In addition, we propose a refinement model EARSB that surgically targets localized generation errors from the output of a prior model. We propose to use a weakly-supervised error classifier to identify localized regions to refine. These identified regions are incorporated into the SB by augmenting its noise schedule with the error classifier's confidence heatmap. Experiments on two benchmark datasets demonstrate that our synthetic data augmentation improves the performance of existing methods and that EARSB enhances the quality of the images generated by prior models.

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
