# APPENDIX

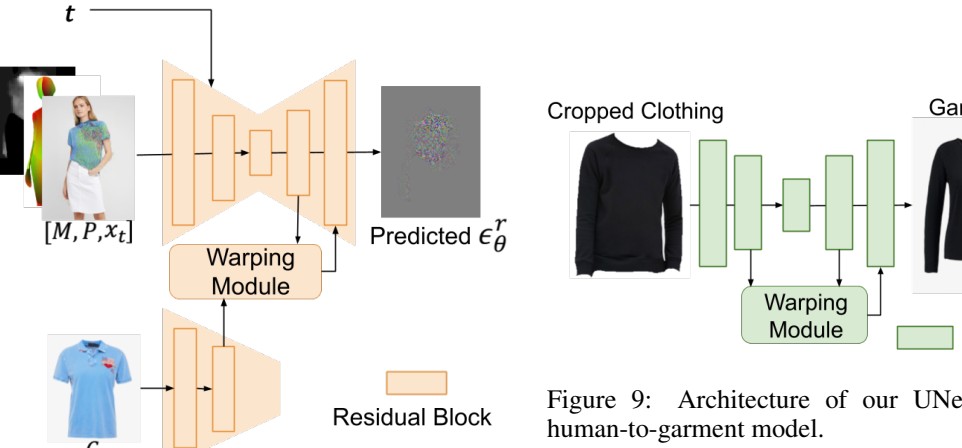

Figure 8: Architecture of our UNet in EARSB.

Figure 9: Architecture of our UNet in the human-to-garment model.

## A IMPLEMENTATIONS DETAILS

For generating the initial image $x_1$ in our EARSB training, we employ two try-on GAN models: HR-VTON (Lee et al., 2022) and SD-VTON (Shim et al., 2024). All human images are processed to maintain their aspect ratio, with the longer side resized to 512 pixels and the shorter side padded with white pixels to reach 512. During training, images undergo random shifting and flipping with a 0.2 probability. The weakly-supervised classifier is trained for 100K iterations with a batch size of 8, while the human-to-garment GAN is trained for 90K iterations with a batch size of 16. EARSB+H2G-UH/FH is trained for 300K iterations with a batch size of 32, incorporating 15% synthetic pairs in each batch. The first 200K iterations are trained on $t \in [0, 1]$ while the following 100k iterations are finetuned on $t \in [0, 0.5)$ and $t \in [0.5, 1]$ respectively following (Balaji et al., 2022). All models utilize the AdamW optimizer with a learning rate of $10^{-4}$.

For inference, we select the GAN model that demonstrates better performance on each dataset to generate the initial image. Specifically, we employ GP-VTON (Xie et al., 2023) for VITON-HD and SD-VTON (Shim et al., 2024) for DressCode-Upper. During the sampling process, the guidance score in Eq. (10) is scaled by a factor of 6 and clamped to the range $[-0.3, 0.3]$.

## B UNET ARCHITECTURE

**EARSB UNet.** The UNet architecture in EARSB consists of residual blocks and garment warping modules. It processes the concatenation of the error map $M$, pose representation $P$, and noisy image $x_t$ to predict the noise distribution $\epsilon_\theta^r$ at time $t$. The UNet encoder has 21 residual blocks, with the number of channels doubling every three blocks to a maximum of 256. Similarly, the garment encoder has 21 residual blocks but reaches a maximum of 128 channels. The decoder mirrors the encoder's structure, with extra garment warping modules. As shown in Fig. 8, each of the first 15 residual blocks in the UNet decoder is followed by a convolutional warping module. These modules concatenate encoded garment features and UNet-decoded features to predict a flow-like map for spatially warping the encoded garment features. The warped features are then injected into the

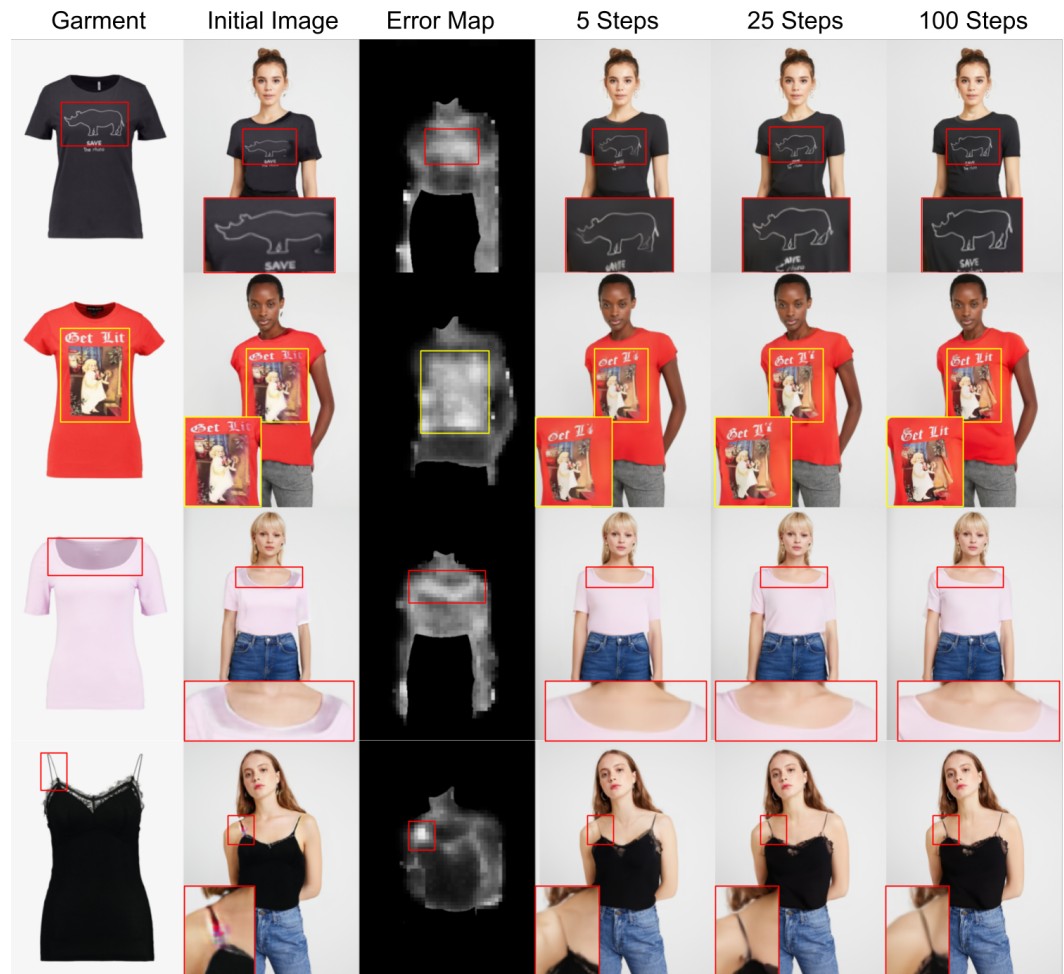

Figure 10: Results on different time steps. Our error map focuses on low-quality regions and maintains the quality of the sufficiently good regions.

subsequent decoder layer via input concatenation. Following (Rombach et al., 2022), all residual blocks and flow-learning modules incorporate timestep embeddings to renormalize latent features.

**Human-to-Garment UNet.** Our human-to-garment UNet architecture is adapted from the model proposed in (Han et al., 2019). As illustrated in Fig. 9, it shares similarities with the UNet in EARSB, but with two key distinctions: a) It is not timestep-dependent and takes cropped clothing as input to generate its product-view image. b) The garment warping module utilizes the $i_{th}$ clothing features from both the encoder and decoder to learn a flow-like map, rather than using encoded features from the human.

## C  VISUALIZING ERROR MAPS

Our EARSB focuses on fixing specific errors and therefore can save the sampling cost when initial predictions are sufficiently good. For example, in the first row of Fig. 10, the error map highlights the graphics and text in the initial image. The highlighted low-quality part is being refined progressively as the number of sampling steps increases from 5 to 100. At the same time, other parts that our weakly-supervised classifier believes to be sufficiently good, which are mostly the solid-color areas, are kept well regardless of the number of sampling steps. Therefore, for an initial image whose error map has almost zero values, we can choose to use fewer steps in sampling. On the contrary, for an initial image whose error map has high confidence, we should assign more sampling steps to it to improve the image quality.

|  | HR-VTON (Lee et al., 2022) | SD-VTON (Shim et al., 2024) | GP-VTON (Xie et al., 2023) |
|---|---|---|---|
| Baseline | 10.75 | 9.05 | 8.61 |
| CAT-DM (Zeng et al., 2024) | 10.03 | 8.76 | 8.55 |
| EARSB | **9.11** | **8.69** | **8.42** |

Table 4: FID scores of using different try-on GAN models to generate the initial image under the unpaired setting.

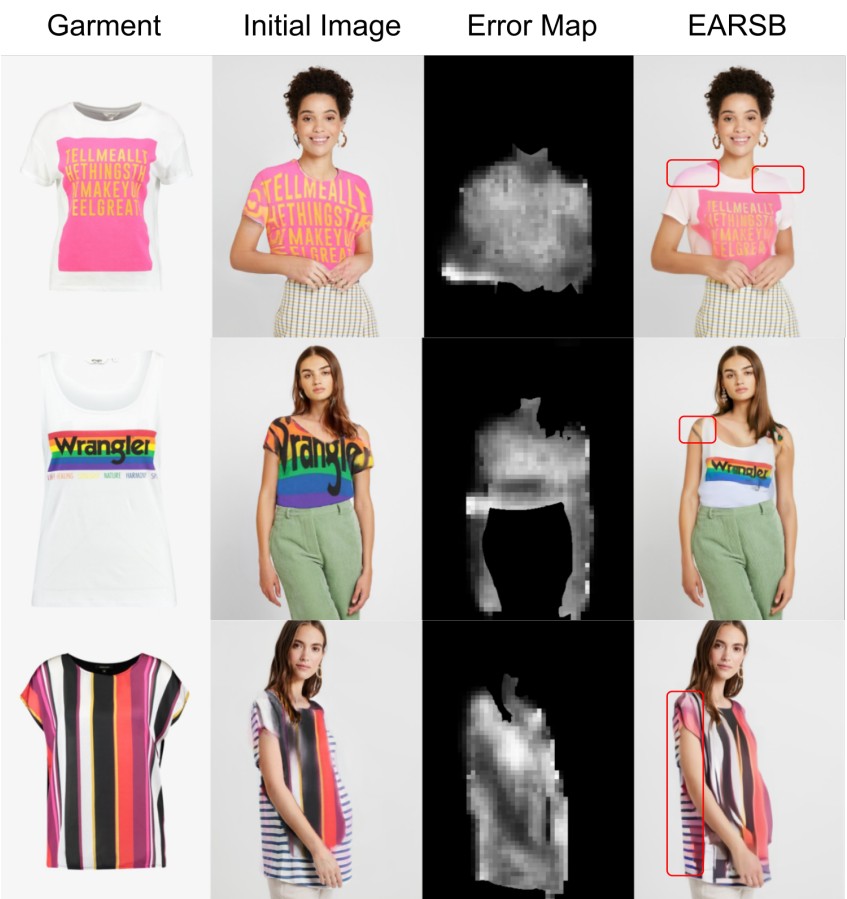

Figure 11: Failure cases on VITON-HD where the initial image has a poor-quality.

## D    ABLATIONS ON THE QUALITY OF THE INITIAL IMAGE $x_1$

In Tab. 4 we include the FID results of using different try-on GAN models to generate the initial image under the unpaired setting. Baseline means the GAN baseline. As previously stated in Sec. 4.1., we can draw three conclusions from the results: a) our EARSB can refine the GAN-generated image over the GAN baseline; b) the quality of the initial image $x_1$ is positively correlated with the quality of the sampled $\hat{x}_0$; c) our model achieves higher gains over CAT-DM, which also tries to refine the GAN-generated image but without error-aware noise schedule.

## E    LIMITATIONS

While our human-to-garment model can effectively generate synthetic paired data for try-on training augmentation, it has some imperfections. The overall quality of synthetic garments is regulated by our filtering criteria (Sec. 3.1.2), yet minor texture deformations occasionally occur. For instance, in Fig. 12, the second pair of the first row shows a misaligned shirt placket in the synthetic garment.

This limitation stems partly from the fact that our model is trained in the image domain which lacks 3D information. A potential solution is to utilize DensePose representations extracted from the garment as in (Cui et al., 2023).

A key constraint of our EARSB is its refinement-based nature, which makes the generated image dependent on the initial image. We assume that the initial image from a try-on GAN model is of reasonable quality, requiring only partial refinement. Consequently, if the initial image is of very poor quality, our refinement process cannot completely erase and regenerate an entirely new, unrelated image. Fig. 11 illustrates this limitation: in the first row, the initial image severely mismatches the white shirt with pink graphics. With EARSB refinement, while the shirt is correctly re-warped, color residuals from the initial image persist around the shoulder area.

## F  ADDITIONAL VISUALIZATIONS

Figures 12 and 13 showcase exemplars from our synthesized datasets H2G-UH and H2G-FH, respectively. The generated garment images closely mimic the product view of the clothing items, accurately capturing both the shape and texture of the original garments worn by the individuals. This approach to creating synthetic training data for the virtual try-on task is both cost-effective and data-efficient, highlighting the benefits of our proposed human-to-garment model.

Figures 14 and 15 give visualized results of the proposed EARSB and EARSB+H2G-UH. In contrast to previous approaches, EARSB specifically targets and enhances low-quality regions in GAN-generated images, which typically correspond to texture-rich areas. This targeted improvement is evident in the last row of Fig. 14, where EARSB more accurately reconstructs text *freinds*, and in the third row, where it successfully generates four side buttons. Furthermore, the incorporation of our synthetic dataset H2G-UH with EARSB leads to even more refined details in the generated images, demonstrating the synergistic effect of our combined approach.

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

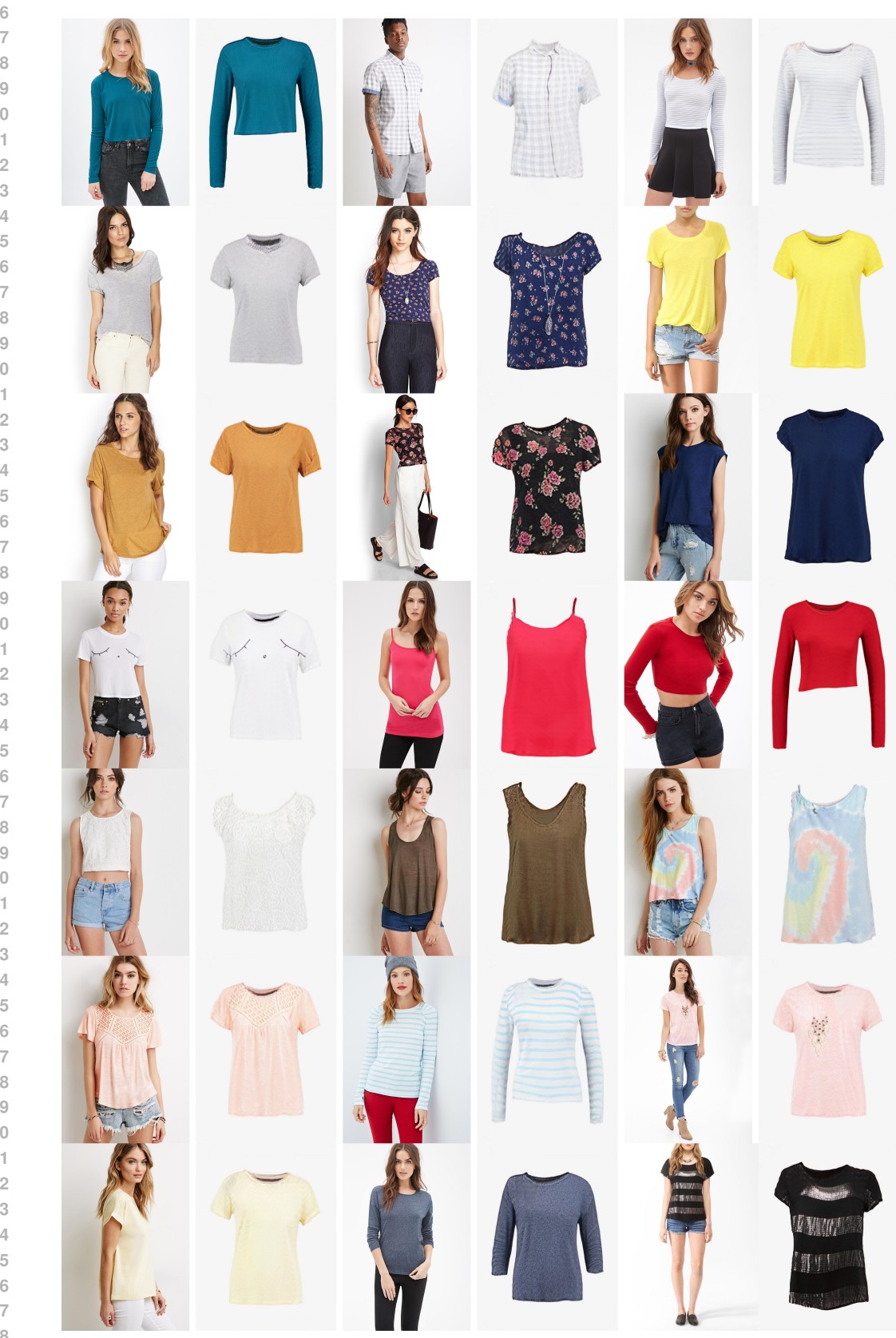

Figure 12: Visualized examples of the (human, synthetic garment) pairs on our proposed H2G-UH.

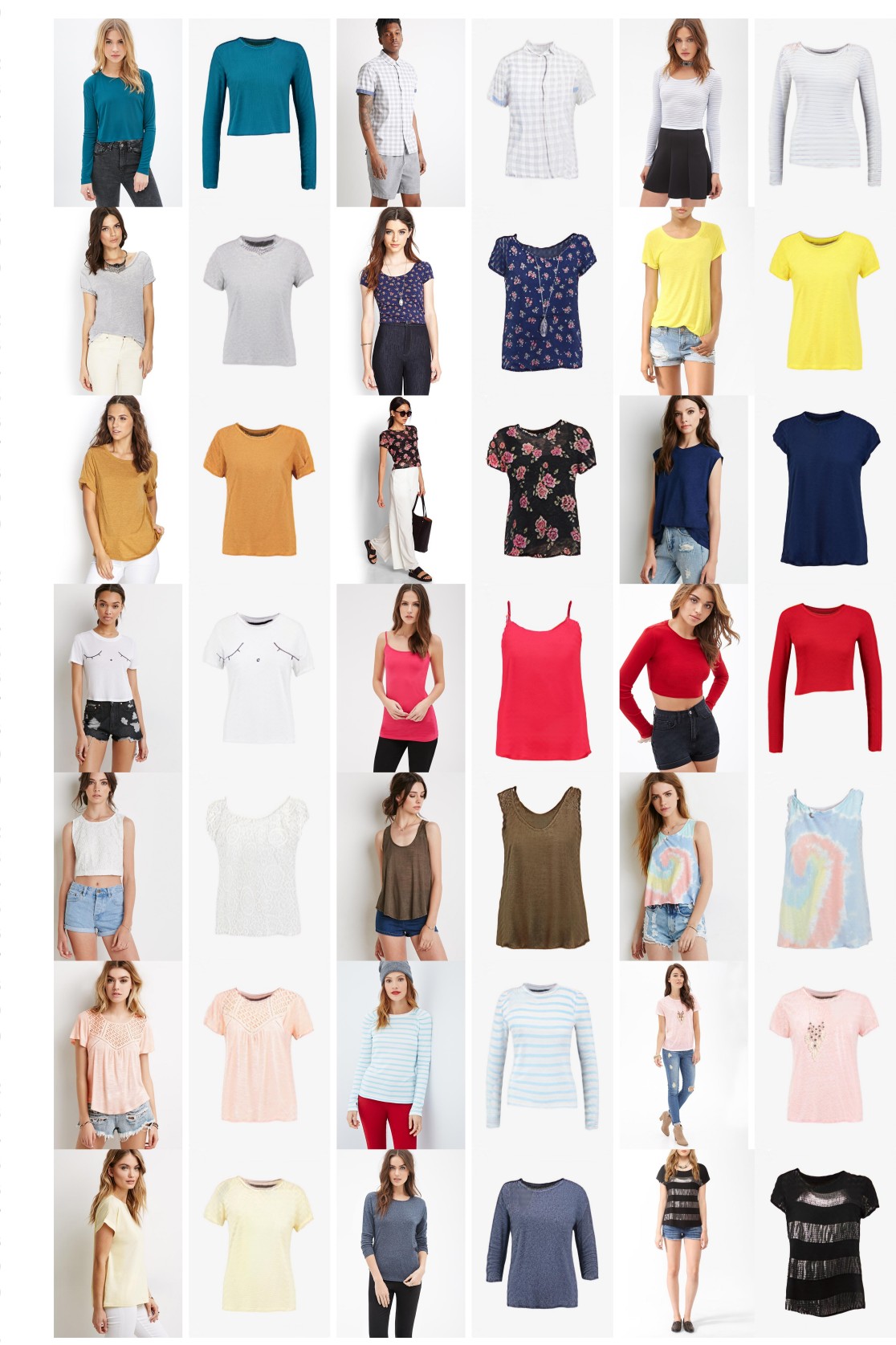

Figure 13: Visualized examples of the (human, synthetic garment) pairs on our proposed H2G-FH.

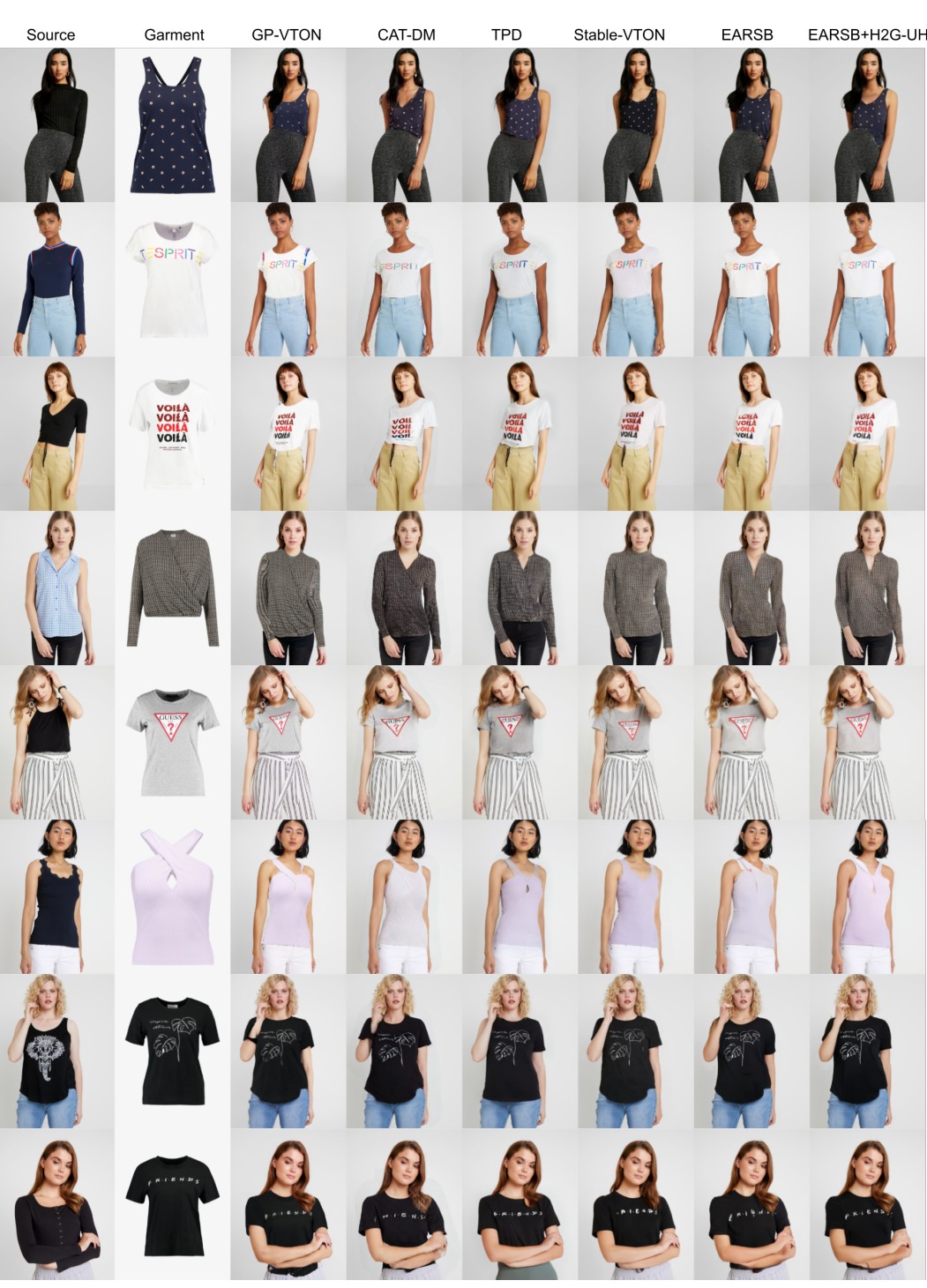

Figure 14: Visualized examples on VITON-HD. Our EARSB and EARSB+H2G-UH better recovers the intricate textures in the garment.

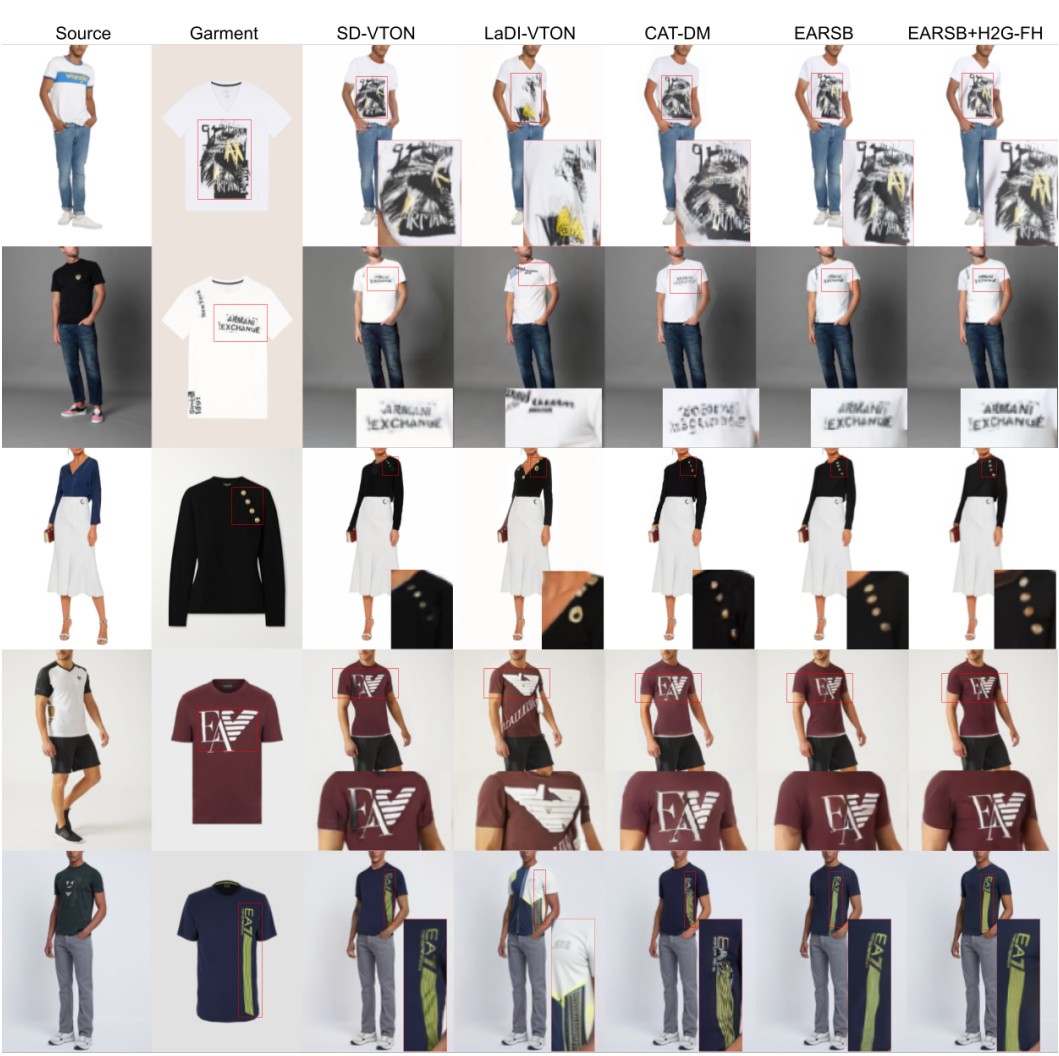

Figure 15: Visualized examples on DressCode-Upper. Our EARSB and EARSB+H2G-UH better reconstructs the texts and graphics in the garment.