# OpenReview forum: "Enhancing Virtual Try-On with Synthetic Pairs and Error-Aware Noise Scheduling"
_ICLR.cc/2025/Conference — ICLR 2025 Conference Withdrawn Submission_

### Official Review · Reviewer_Uiak · 2024-10-27

**Soundness:** 2
**Presentation:** 3
**Contribution:** 2
**Rating:** 5
**Confidence:** 4

**Summary:**

This paper introduces two methods to enhance virtual try-on performance. The first method utilizes a GAN-based approach to generate synthetic data for training try-on models, creating garment data from single human images drawn from multiple datasets. The second method focuses on training the try-on model itself, which again consists of two stages. The first stage is a GAN-based method that produces intermediate results, while the second stage is an image-to-image model based on I2SB, which refines the initial outputs. Unlike the naive I2SB approach, the proposed method incorporates an error map module to emphasize low-quality regions.

**Strengths:**

1. The paper is well-structured and easy to follow, with clear presentation.
2. The proposed method outperforms several baseline methods in virtual try-on tasks.
3. The synthetic data generation approach could be beneficial for future research.

**Weaknesses:**

1. The paper does not offer significant technical contributions, primarily combining existing methods, such as combining GAN outputs with I2SB for post-processing.
2. According to Table 3, combining existing methods with synthetic data sometimes yields better performance than the proposed method.
3. Minor formatting issues. For example, some equations (e.g., Eq 2 and Eq 8) are missing proper punctuation.
4. Inconsistency in terminology, with both "error-map" and "error map" used throughout the paper.

**Questions:**

1. I feel the training process of EARSB is not clearly explained. How are data pairs prepared for this stage? What initialization is used for the U-Net, and what is the training loss?
2. In I2SB, training assumes that the posterior, given a boundary pair, follows a Gaussian distribution. Does WSC affect the Gaussian properties?

**Details Of Ethics Concerns:**

N.A.

---

> ### Author Response · Authors · 2024-11-15
>
> 1. The paper does not offer significant technical contributions, primarily combining existing methods, such as combining GAN outputs with I2SB for post-processing.
>
> The proposed EARSB is a training-based refinement method that learns to refine the artifacts of an initial image. The major contributions of our work are two folds: a) from the data perspective, we propose a synthetic data augmentation strategy to improve virtual try-on models; b) from the model perspective, we introduce **spatially adaptive** noise schedule in the Schr\"odinger Bridge, which improves the low-quality region of an initially generated image based on a spatially-varying noise schedule that targets known artifacts.
>
> 2. According to Table 3, combining existing methods with synthetic data sometimes yields better performance than the proposed method.
>
> The superior performance of Stable-VTON+H2G-UH in Table 3 is partially attributed to the powerful generation ability of Stable Diffusion backbone, which is pretrained on millions of images. We updated Table 1 by using the SD-based model CatVTON to generate the initial image in EARSB(SD)+H2G-UH/FH. See the comments posted for reviewer RZB7 for the updated Table 1. It is shown that EARSB(SD)+H2G-UH/FH gains over the SD-based methods, including Stable-VTON+H2G-UH.
>
> 3. Minor formatting issues. For example, some equations (e.g., Eq 2 and Eq 8) are missing proper punctuation.
>
> Thanks for pointing out the typos. We've corrected them in the revised draft.
>
> 4. Inconsistency in terminology, with both "error-map" and "error map" used throughout the paper.
>
> We use "error-map" as a compound modifier before a noun (e.g., error-map-reweighted noise schedule) and "error map" as a noun phrase.
>
> 5. I feel the training process of EARSB is not clearly explained. How are data pairs prepared for this stage? What initialization is used for the U-Net, and what is the training loss?
>
> The training of EARSB is conducted in three steps: 1) Preprocess the images in the training set and feed the preprocessed inputs to existing try-on models to get the initial images $x_1$; 2) Obtain the error maps $M$ on the initial images $x_1$ using our WSC. 3) Use $M$ to reweight the noise schedule in I$^2$SB and train the noise prediction UNet in EARSB following Eq. (9). We have clarified the above procedure in the revised draft.
>
> We use Xavior initialization for the UNet. Following prior diffusion models, the training loss is a L2 loss between the predicted noise and the true noise. More details are included in the appendix.
>
> 6. In I2SB, training assumes that the posterior, given a boundary pair, follows a Gaussian distribution. Does WSC affect the Gaussian properties?
>
> WSC does not affect the Gaussian properties because the predicted error map acts as a multiplier to the Gaussian noise in I2SB, which only affects its variance.

---

### Official Review · Reviewer_RZB7 · 2024-10-31

**Soundness:** 4
**Presentation:** 3
**Contribution:** 3
**Rating:** 5
**Confidence:** 4

**Summary:**

This paper aims to address two main challenges in the virtual try-on field:

1. Difficulty in obtaining paired data (i.e., garment images and images of people wearing the corresponding garment) for use in datasets.
2. Existing virtual try-on models struggle to generate accurate texture details and patterns of the garment.

To tackle the first issue, this paper proposes a model that generates corresponding garment images based on images of people wearing the garment. The generated garment images and original people images are used to train virtual try-on models.

For the second issue, the paper employs a weakly supervised classifier to generate an error map, allowing the model to identify regions that need refinement, which is achieved using a Schrödinger bridge.

**Strengths:**

1. Extensive experiments demonstrate the effectiveness of the method proposed in this paper.
2. The data augmentation approach introduced in this paper can be easily applied to other virtual try-on models, and this paper have verified this on both Stable-VITON and CAT-DM.
3. Compared to adding random noise directly to GAN-generated images in CAT-DM, using a Schrödinger bridge to refine erroneous areas appears to be more effective.

**Weaknesses:**

1. The performance improvement of the model does not seem very significant; as shown in Table 3, Stable-VTON + H2G-UH appears to achieve better performance. Perhaps the single U-Net structure limits the model’s generative capability? However, as stated in this paper, SD-based methods like Stable-VTON use pretrained weights, so this is not a major issue.
2. The baselines compared in this paper are somewhat outdated; more recent virtual try-on models such as IDM-VTON [1], OOTDiffusion [2], and CATVTON [3] should perhaps be included in the comparison experiments.
3. The results of LaDI-VTON on VITON-HD and Stable-VITON on DressCode-Upper are missing in Table 1.
4. Although there are some visualized examples of the synthetic garment images, I am somewhat concerned about the quality of the generated images. Could you provide some quantitative result to support this?
5. Although this paper compares the results of different try-on GAN models for generating the initial image and demonstrates that EARSB can refine GAN-generated images, I am curious whether EARSB could have a similar effect on diffusion-generated images as well.

[1] Choi Y, Kwak S, Lee K, et al. Improving diffusion models for virtual try-on[J]. arXiv preprint arXiv:2403.05139, 2024.

[2] Xu Y, Gu T, Chen W, et al. Ootdiffusion: Outfitting fusion based latent diffusion for controllable virtual try-on[J]. arXiv preprint arXiv:2403.01779, 2024.

[3] Chong Z, Dong X, Li H, et al. CatVTON: Concatenation Is All You Need for Virtual Try-On with Diffusion Models[J]. arXiv preprint arXiv:2407.15886, 2024.

**Questions:**

1. To obtain the error map, this paper manually annotated 5% of the GAN-generated images. Could you analyze the impact of the annotation ratio on model performance?
2. I would like to confirm which dataset the human-to-garment model was trained on. Was it VITON-HD or DressCode?

---

> ### Author Response · Authors · 2024-11-15
>
> We thank the reviewer for the informative feedback and address the concerns below:
>
> 1. IDM-VTON, OOTDiffusion, and CATVTON should perhaps be included in the comparison experiments.
>
> IDM-VTON from ECCV 2024 was published at the time of the submission. OOTDiffusion is an unpublished work and CATVTON is a concurrent work. To compare with the most recent model that shows better performance, we use CatVTON, which is a concurrent work, as one of our baselines. The experiment section and Table 1 have been updated accordingly as below:
>
> | | VITON-HD | | | | | | DressCode-Upper | | | | | |
> |---|:---|---|:---|---|---|---|:---|---|:---|---|---|---|
> | | Unpaired | | Paired | | | | Unpaired | | Paired | | | |
> | | FID↓ | KID↓ | FID↓ | KID↓ | SSIM↑ | LPIPS↓ | FID↓ | KID↓ | FID↓ | KID↓ | SSIM↑ | LPIPS↓ |
> | **GAN-Based** | | | | | | | | | | | | |
> | GP-VTON | 8.61 | 0.86 | 5.53 | 0.07 | 0.913 | 0.064 | 26.19 | 1.71 | 23.66 | 1.59 | 0.816 | 0.262 |
> | **SD-Based** | | | | | | | | | | | | |
> | LaDI-VTON | 8.95 | 0.12 | 6.05 | 0.08 | 0.902 | 0.071 | 14.88 | 0.39 | 11.61 | 0.32 | 0.939 | 0.057 |
> | TPD | 8.23 | **0.06** | **4.86** | 0.04 | 0.917 | 0.057 | - | - | - | - | - | - |
> | Stable-VTON | 8.20 | 0.07 | 5.16 | 0.05 | 0.917 | 0.057 | - | - | - | - | - | - |
> | IDM-VTON | 8.59 | 0.11 | 5.51 | 0.09 | 0.902 | 0.061 | 11.09 | 0.16 | 6.79 | 0.12 | 0.956 | 0.026
> | CatVTON | 8.87 | 0.08 | 5.49 | 0.07 | 0.915 | 0.059 | 11.91 | 0.21 | 7.66 | 0.10 | 0.950 | 0.038 |
> | EARSB | 8.42 | 0.07 | 5.25 | 0.05 | 0.918 | 0.059 | 10.89 | 0.13 | 7.15 | 0.13 | 0.961 | 0.028 |
> | EARSB +H2G-UH/FH | 8.26 | **0.06** | 5.14 | 0.04 | 0.919 | 0.058 | 10.70 | 0.11 | 7.05 | 0.11 | 0.965 | 0.026 |
> | EARSB(SD) +H2G-UH/FH | **8.04** | **0.06** | 4.90 | **0.03** | **0.925** | **0.053** | **10.41** | **0.09** | **6.76** | **0.08** | **0.968** | **0.023** |
>
> In the updated table, EARSB(SD)+H2G-UH/FH uses diffusion model CatVTON to generate the initial image. In summary, EARSB and EARSB+H2G-UH/FH improve the GAN baselines while EARSB(SD)+H2G-UH/FH gains over the SD-based methods. Overall, our refinement model improves the metrics of existing methods on both datasets, with further improvements from incorporating synthetic training pairs.
>
> 2. The results of LaDI-VTON on VITON-HD and Stable-VITON on DressCode-Upper are missing in Table 1.
>
> Thanks for catching this! The results of LaDI-VTON on VITON-HD are now included in Table 1 as the above. As for Stable-VITON, the authors have not released their model checkpoint on DressCode-Upper. We require all diffusion models to be evaluated using the same number of sampling steps for fair comparison. Therefore, their results on DressCode-Upper will remain blank in Table 1.
>
> 3. Although there are some visualized examples of the synthetic garment images, I am somewhat concerned about the quality of the generated images. Could you provide some quantitative result to support this?
>
> Yes, we compute the FID, KID, SSIM and LPIPS of the generated garment on the paired test set of VITON-HD and DressCode-Upper, respectively. Results are reported under 1024x1024 image resolution:
>
> | | FID | KID | SSIM | LPIPS |
> |---|---|---|---|---|
> | VITON-HD | 14.81 | 0.42 | 0.849 | 0.229 |
> | DressCode-Upper | 18.92 | 0.59 | 0.832 | 0.257 |
>
>
> 4. I am curious whether EARSB could have a similar effect on diffusion-generated images as well.
>
> In the updated Table 1, we use the diffusion model CatVTON to generate the initial image in EARSB(SD)+H2G-UH/FH. It is shown that EARSB(SD)+H2G-UH/FH gains over the SD-based methods, which is a consistent conclusion we draw from using GAN-generated image as the initial image. Note that while EARSB(SD)+H2G-UH/FH shows better performance, it also brings extra computational costs for obtaining the initial image.
>
> 5. Could you analyze the impact of the annotation ratio on model performance?
>
> The reason for annotating a small portion of the generated images is to get reasonable detection results of the artifacts while minimizing manual labeling efforts. It is expected to get higher detection accuracy with higher labeling costs and a higher annotation ratio. 5\% might not be the optimal ratio to balance the detection accuracy and the labeling efforts, but it is cost-effective in making the classifier outperform self-supervised and unsupervised training methods, as illustrated in Figure 7. The refinement results in Table 1 also demonstrate that it is effective in improving existing methods. In future work, following the data scaling law, we will explore the relationship between the labeling ratio, the time spent on annotating the data, and the final model performance.
>
> 6. I would like to confirm which dataset the human-to-garment model was trained on. Was it VITON-HD or DressCode?
>
> For H2G-UH which contains mostly upper-body human images, we train the human-to-garment model on VITON-HD and for H2G-FH which contains mostly full-body human images, the human-to-garment model is trained on DressCode-Upper.

---

### Official Review · Reviewer_JB7X · 2024-11-04

**Soundness:** 3
**Presentation:** 3
**Contribution:** 2
**Rating:** 5
**Confidence:** 5

**Summary:**

This paper explores virtual fitting techniques that aim to demonstrate the effect of a person wearing a specific garment by generating new images. The authors propose a garment extraction model that generates (person, synthetic garment) pairs from a single image of a wearing individual, which helps to address the problem of insufficient training data. An Error-Aware Refinement-based Schrodinger Bridge(EARSB) is proposed, which is capable of targeting and correcting local generation errors in the output of the virtual try-on model. This error-aware noise scheduling-based approach is an innovative technical tool.

**Strengths:**

+ 1. The abstract and introduction sections clearly outline the motivation, methodology, and contributions of the study. The methodology section provides a detailed description of the technical details.

+ 2. The paper employs an advanced SB method for denoising training, which can better accomplish personalized design tailored for the VTON task.

+ 3. The author compares the performance of EARSB with existing methods and discusses the impact of synthetic data augmentation, which helps demonstrate the effectiveness of EARSB.

**Weaknesses:**

- 1. The error map-reweighted diffusion process appears to be a relatively universal technique, but the author has not provided further introductions to its applications in this section. Therefore, there is no way to prove whether this method can be applied to other methods and bring about similar performance improvements.

- 2. More visualizations are needed to show how good regions are assigned less noise and bad regions are assigned more noise. Simply relying on Figure 4 does not adequately convey the precise meaning of this innovation.

- 3. The Diffusion-VTON series of methods lacks comparisons with some necessary approaches, such as quantitative and qualitative comparisons with IDM-VTON, DCI-VTON, and other methods.

- 4. The explanation of the EARSB model in the paper is not sufficient, especially regarding the details of how the model identifies and corrects local errors. Can the optimization of local details be achieved solely through weakly supervised masks?

- 5. The paper mentions that SD-based methods require large-scale pretraining, while EARSB does not. How does EARSB ensure the effectiveness and generalization ability of VTON in scenarios beyond the dataset?

-6. I did not find more introductions related to the code of the paper, which reduces the reproducibility of this paper.

**Questions:**

Please refer to the "Weaknesses."

---

> ### Author Response · Authors · 2024-11-15
>
> We thank the reviewer for the informative feedback and we address the concerns as follows:
>
> 1. There is no way to prove whether EARSB can be applied to other methods and bring about similar performance improvements.
>
> The proposed synthetic data augmentation strategy has been proven to be beneficial when added to other try-on models. At the same time, the introduced EARSB model architecture is specific for the virtual try-on task and might need major architectural changes to be adapted to other vision tasks. We mainly focus on virtual try-on under the scope of this paper and therefore did not test EARSB on other applications. But, the refinement-based idea in EARSB still applies to the general setting where we want to refine a flawed image output, as suggested by reviewer d9nt. In future work, we could expand the scope and apply EARSB to other image editing applications.
>
> 2. More visualizations are needed to show how good regions are assigned less noise and bad regions are assigned more noise. Simply relying on Figure 4 does not adequately convey the precise meaning of this innovation.
>
> More visualized examples of the error map can be found in Figure 10  of the Appendix. The error map reweights the noise schedule such that good regions are assigned less noise and bad regions are assigned more noise. More precisely, we multiply the noise distribution $\epsilon$ with the error map $M$ as in Eq. (4). As a result, a higher volume of noise is assigned to the low-quality regions so the model can focus on refining these regions. The intuition behind this is that good regions need less modification learned from the diffusion process while bad regions need more. We have replaced Figures 3 and 4 with another image example to better clarify the benefit of the noise reweighting.
>
> 3. The Diffusion-VTON series of methods lacks comparisons with some necessary approaches, such as quantitative and qualitative comparisons with IDM-VTON, DCI-VTON, and other methods.
>
> IDM-VTON from ECCV 2024 was published at the time of the submission and DCI-VTON is an earlier method published in ACM MM 2023. To compare with the most recent model that shows better performance, we use CatVTON, which is a concurrent work, as one of our baselines. The experiment section and Table 1 have been updated accordingly. See the comments posted for reviewer RZB7 for the updated Table 1. In the updated table, EARSB(SD)+H2G-UH/FH uses diffusion model CatVTON to generate the initial image. In summary, EARSB and EARSB+H2G-UH/FH improve the GAN baselines while EARSB(SD)+H2G-UH/FH gains over the SD-based methods. Overall, our refinement model improves the metrics of existing methods on both datasets, with further improvements from incorporating synthetic training pairs.
>
> 4. The explanation of the EARSB model in the paper is not sufficient, especially regarding the details of how the model identifies and corrects local errors. Can the optimization of local details be achieved solely through weakly supervised masks?
>
> We have revised the draft in Section 3.2 to clarify our model EARSB. Our model identifies the local errors in the error map predicted by the weakly-supervised classifier and refines these errors via the Schr\"odinger Bridge sampling with spatially adaptive noise schedule reweighted by the error map. Technically, the local refinement can be solely achieved by inpainting the erroneous regions in the weakly supervised masks. However, learning the error-to-ground-truth mapping in our EARSB will provide more explicit guidance on how to map specific errors for refinement.
>
> We updated Table 3(c) with the above inpainting method, denoted as Inpaint below. Inpaint's degraded performance demonstrates the importance of learning the error-to-ground-truth refinement in our model.
> | | Unpaired | Paired | | |
> | --- | :---: | :---: | :---: | :---: |
> | | FID↓ | FID↓ | SSIM↑ | LPIPS↓ |
> | Inpaint | 9.26 | 6.33 | 0.909 | 0.068 |
> | EARSB | 8.42 | 5.25 | 0.918 | 0.059 |
>
> 5. The paper mentions that SD-based methods require large-scale pretraining, while EARSB does not. How does EARSB ensure the effectiveness and generalization ability of VTON in scenarios beyond the dataset?
>
> From the perspective of a fair comparison, directly comparing EARSB and SD-based models is less fair because SD-based methods are trained on large-scale data. However, since EARSB is a refinement-based model, we can use images generated by SD-based methods as the initial image and refine their clothing textures in EARSB. This is an implicit approach to ensure the generalization ability of EARSB. We've updated Table 1 to include this approach, denoted by EARSB(SD)+H2G-UH/FH. See the comments posted for reviewer RZB7 for the updated Table 1. In the updated table, EARSB(SD)+H2G-UH/FH uses diffusion model CatVTON to generate the initial image, showing performance gains over the SD-based methods.

---

### Official Review · Reviewer_d9nt · 2024-11-04

**Soundness:** 3
**Presentation:** 3
**Contribution:** 2
**Rating:** 5
**Confidence:** 4

**Summary:**

The authors propose a framework for improved generation of virtual try-on images by leveraging synthetic data and an error correction module. The synthetic data is created from images of clothed individuals by using a garment extraction module. These synthetic pairs are then used to as augmented data during training of the virtual try on module. The authors also propose error aware refinement based Schrödinger bridge that aim to correct localized errors in the generated image. State of the art performance is demonstrated with a 59% user preference for try-on results generated by the proposed approach.

**Strengths:**

1. **Claity**: The paper is well written with adequate details and explanation provided for all design components. In particular, the specific details regarding the training of garment extraction network and error detection network is helpful to reproduce the approach.
2. **Comparisions**: The framework shows state of the art quantitative performance compared to a number of recent baselines. The user study additionally strengths the claim of user preference and highlights the efficacy of the proposed approach.
3. **Ablation**: A number of detailed ablations are provided to highlight the need and effect of data augmentation and the use of manually annotated data for training the error detection network.
4. **Garment-Extraction network**: The idea of reducing the task of virtual try-on to a simpler problem by training a network to generate garments in canonical pose for an input dressed individual in any arbitrary pose is an interesting way of obtaining useful training data.
5. **Error-detection network**: Using manually annotated data to train a network to recognize error regions, and then using the detected error regions to encourage local refinement is an interesting idea.
6. **Result Quality**: A number of qualitative results are provided on a wide variety of example that demonstrates that high frequency textures and details of lines/ logos are maintained better with the proposed approach.

**Weaknesses:**

1. **Novelty**: Although the authors present an interesting approach to improve the quality of generated virtual try on images, the novelty is somewhat limited to the synthetic pair generation component. In particular, the other components such as $I^2SB$ have been previously introduced and have only been adapted to this framework. Highlighting the main novel components and differences from prior work would aid in addressing this concern.
2. **Relevance**: Although virtual try-on represents an important problem in the space of computer vision, the findings in the paper are primarily described in context of improving quality for try on task. Briefly commenting about how some of these components can be adapted to other general settings would be helpful. In particular, the finding that 5% of generated images can be manually annotated to identify error prone areas and used to train a error prediction network is potentially useful in many pipelines that would use a refinement network.
3. **Details about the manual annotation**: The authors mention that 5% of generated images are manually annotated with error regions. Providing a small ablation showing the effect of training the network with lesser and more annotated data would be helpful to highlight the importance of this finding.
4. **Target clothing shape**: Although the generated images maintain the texture of the target clothing item, most times the generated image does not fully respect the shape of the target clothing item (eg. ll345-349 makes the dress longer than the product, ll328-333 makes the generated clothing shorter than the product. The results mostly appear like the texture of the target clothing is transferred in detail into the shape of the source posed image. Providing more examples of (short source, long target) or (long source, short target) pairs, would be helpful to understand its limitations and generalizability.

**Questions:**

1. The authors use a GAN based model for the first stage and then use a Schrödinger bridge for the refinement network. What is the motivation behind using a GAN based base network as opposed to a diffusion based base network? (particularly in light of the fact that the next best performant networks are SD based). Providing a brief explanation regarding the choice of base network would be insightful.
2. How many total number of images are 5% of generated images?

---

> ### Author Response · Authors · 2024-11-15
>
> We thank the reviewer for the informative feedback and we address the concerns as follows:
>
> 1. **Novelty**
>
> The major contributions of our work are two folds: a) from the data perspective, we propose a synthetic data augmentation strategy to improve virtual try-on models; b) from the model perspective, the major difference between the proposed EARSB and prior diffusion models is the introduced spatially adaptive noise schedule, which improves the low-quality region of an initially generated image based on a spatially-varying noise schedule that targets known artifacts. To clarify the second contribution, we have highlighted the spatially adaptive noise schedule in the introduction section of the revised draft.
>
> 2. **Relevance**
>
> Mentioning that our weakly-supervised classifier can be adapted to other tasks is a great suggestion. We added the following description in the first paragraph of Section 3.4 : "This refinement goal applies to the general setting where we want to refine a flawed image output. In the scope of this paper, we focus on the virtual try-on task. "
>
>
> 3. **Details about the manual annotation**
>
> The reason for annotating a small portion of the generated images is to get reasonable detection results of the artifacts while minimizing manual labeling efforts. It is expected to get higher detection accuracy with higher labeling costs and a higher annotation ratio. 5\% might not be the optimal ratio to balance the detection accuracy and the labeling efforts, but it is cost-effective in making the classifier outperform self-supervised and unsupervised training methods, as illustrated in Figure 7. The refinement results in Table 1 also demonstrate that it is effective in help improving existing try-on methods. In future work, following the data scaling law, we will explore the relationship between the labeling ratio, the time spent on annotating the data, and the final model performance.
>
> 4. **Target clothing shape**
>
> The shape of the upper clothing when worn by an individual is primarily decided by two factors: a) the physical interactions between the upper clothing and the lower clothing; b) the size match between the garment and the individual. For a, n example is that the tank top in ll328-333 becomes shorter on the human because it is tucked into the pants. For b, an example is that the same outfit could look loose or tight on different people depending on their body shape. These are difficult to solve in the 2D domain without knowing the garment or body measurement data. Some prior work addressed this problem in a human-interactive setting [1,2]. In this paper, we focus on non-interactive virtual try-on and only aim to transfer the texture of the garment to the person in a reasonable shape. Note that our EARSB is a refinement-based model that targets known artifacts. Therefore, it is expected to follow the shape of the clothing in the initial image unless it is misaligned with the pose or has other artifacts around the boundaries. In future work, we might explore incorporating human interactions into the model to control the dressing style and clothing length.
>
> [1] Chen, C. Y., Chen, Y. C., Shuai, H. H., Cheng, W. H. Size does matter: Size-aware virtual try-on via clothing-oriented transformation try-on network. In ICCV 2023.
>
> [2] Pan X, Tewari A, Leimkühler T, Liu L, Meka A, Theobalt C. Drag your gan: Interactive point-based manipulation on the generative image manifold. In ACM SIGGRAPH 2023.
>
> 5. What is the motivation behind using a GAN based base network as opposed to a diffusion based base network?
>
> This is because generating the initial image using a GAN model is cheaper as the diffusion-based model has a notoriously low sampling speed. To test EARSB's performance without considering the sampling cost, we use the diffusion model CatVTON to generate the initial image and refine it with our EARSB. We have updated Table 1 with the new results, denoted as EARSB(SD) +H2G-UH/FH.  See the comments posted for reviewer RZB7 for the updated Table 1. EARSB and EARSB+H2G-UH/FH improve the GAN baselines while EARSB(SD)+H2G-UH/FH gains over the SD-based methods. Overall, our refinement model improves the metrics of existing methods on both datasets, with further improvements from incorporating synthetic training pairs.
>
> 6. How many total number of images are 5\% of generated images?
>
> We use three GAN models to generate the initial image in training, which makes the total number of annotations 3*0.05*11647=1747 on VITON-HD and  3*0.05*13564=2034 on DressCode-Upper.

---

### Note · Authors · 2024-11-15

I have read and agree with the venue's withdrawal policy on behalf of myself and my co-authors.